# Function-Space Distributions over Kernels

**Gregory W. Benton**[*1]    **Wesley J. Maddox**[*2]    **Jayson P. Salkey**[*1]
**Júlio Albinati**[‡3]    **Andrew Gordon Wilson**[1,2]

[1]Courant Institute of Mathematical Sciences, New York University
[2]Center for Data Science, New York University
[3]Microsoft

## Abstract

Gaussian processes are flexible function approximators, with inductive biases controlled by a covariance kernel. Learning the kernel is the key to representation learning and strong predictive performance. In this paper, we develop *functional kernel learning* (FKL) to directly infer functional posteriors over kernels. In particular, we place a transformed Gaussian process over a spectral density, to induce a non-parametric distribution over kernel functions. The resulting approach enables learning of rich representations, with support for any stationary kernel, uncertainty over the values of the kernel, and an interpretable specification of a prior directly over kernels, without requiring sophisticated initialization or manual intervention. We perform inference through elliptical slice sampling, which is especially well suited to marginalizing posteriors with the strongly correlated priors typical to function space modeling. We develop our approach for non-uniform, large-scale, multi-task, and multidimensional data, and show promising performance in a wide range of settings, including interpolation, extrapolation, and kernel recovery experiments.

## 1   Introduction

Practitioners typically follow a two-step modeling procedure: (1) choosing the functional form of a model, such as a neural network; (2) focusing learning efforts on training the parameters of that model. While inference of these parameters consume our efforts, they are rarely interpretable, and are only of interest insomuch as they combine with the functional form of the model to make predictions. Gaussian processes (GPs) provide an alternative *function space* approach to machine learning, directly placing a distribution over functions that could fit data [25]. This approach enables great flexibility, and also provides a compelling framework for controlling the inductive biases of the model, such as whether we expect the solutions to be smooth, periodic, or have conditional independence properties.

These inductive biases, and thus the generalization properties of the GP, are determined by a kernel function. The performance of the GP, and what representations it can learn, therefore crucially depend on what we can learn about the kernel function itself. Accordingly, kernel functions are becoming increasingly expressive and parametrized [15, 31, 34]. There is, however, no a priori reason to assume that the true data generating process is driven by a particular parametric family of kernels.

We propose extending the function-space view to kernel learning itself – to represent uncertainty over the kernel function, and to reflect the belief that the kernel does not have a simple parametric form. Just as one uses GPs to directly specify a prior and infer a posterior over functions that can fit data, we propose to directly reason about priors and posteriors over kernels. In Figure 1, we illustrate the shift from standard function-space GP regression, to a function-space view of kernel learning.

Specifically, our contributions are as follows:

---

[*]Equal contribution. [‡]Work done while interning with AGW.

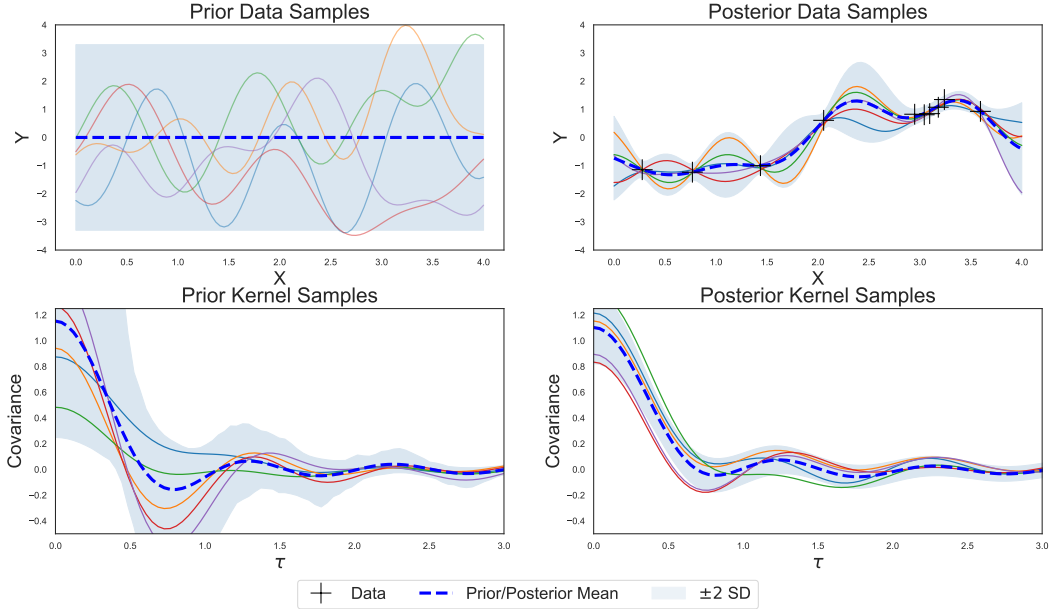

Figure 1: **Above:** A function-space view of regression on data. We show draws from a GP prior and posterior over functions in the left and right panels, respectively. **Below:** With FKL, we apply the function-space view to *kernels*, showing prior kernel draws on the left, and posterior kernel draws on the right. In both cases, prior and posterior means are in thick black, two standard deviations about the mean in grey shade, and data points given by crosses. With FKL, one can specify the prior mean over kernels to be any parametric family, such an RBF kernel, to provide a useful *inductive bias*, while still containing support for *any* stationary kernel.

- We model a spectral density as a transformed Gaussian process, providing a non-parametric function-space distribution over kernels. Our approach, *functional kernel learning* (FKL), has several key properties: (1) it is highly flexible, with support for any stationary covariance function; (2) it naturally represents uncertainty over all values of the kernel; (3) it can easily be used to incorporate intuitions about what types of kernels are *a priori* likely; (4) despite its flexibility, it does not require sophisticated initialization or manual intervention; (5) it provides a conceptually appealing approach to kernel learning, where we reason directly about prior and posterior kernels, rather than about parameters of these kernels.

- We further develop FKL to handle multidimensional and irregularly spaced data, and multi-task learning.

- We demonstrate the effectiveness of FKL in a wide range of settings, including interpolation, extrapolation, and kernel recovery experiments, demonstrating strong performance compared to state-of-the-art methods.

- Code is available at `https://github.com/wjmaddox/spectralgp`.

Our work is intended as a step towards developing Gaussian processes for *representation learning*. By pursuing a function-space approach to kernel learning, we can discover rich representations of data, enabling strong predictive performance, and new interpretable insights into our modeling problems.

## 2   Related Work

We assume some familiarity with Gaussian processes [e.g., 25]. A vast majority of kernels and kernel learning methods are parametric. Popular kernels include the parametric RBF, Matérn, and periodic kernels. The standard multiple kernel learning [11, 12, 16, 24] approaches typically involve additive compositions of RBF kernels with different bandwidths. More recent methods model the spectral density (the Fourier transform) of stationary kernels to construct kernel learning procedures. Lázaro-Gredilla et al. [17] models the spectrum as independent point masses. Wilson and Adams [34] models the spectrum as a scale-location mixture of Gaussians, referred to as a *spectral mixture*

*kernel* (SM). Yang et al. [39] combine these approaches, using a random feature expansion for a spectral mixture kernel, for scalability. Oliva et al. [23] consider a Bayesian non-parametric extension of Yang et al. [39], using a random feature expansion for a Dirichlet process mixture. Alternatively, Jang et al. [15] model the parameters of a SM kernel with prior distributions, and infer the number of mixture components. While these approaches provide strong performance improvements over standard kernels, they often struggle with difficulty specifying a prior expectation over the value of the kernel, and multi-modal learning objectives, requiring sophisticated manual intervention and initialization procedures [13].

A small collection of pioneering works [30, 31, 38] have considered various approaches to modeling the spectral density of a kernel with a Gaussian process. Unlike FKL, these methods are constrained to one-dimensional time series, and still require significant intervention to achieve strong performance, such as choices of windows for convolutional kernels. Moreover, we demonstrate that even in this constrained setting, FKL provides improved performance over these state-of-the-art methods.

## 3 Functional Kernel Learning

In this section, we introduce the prior model for *functional kernel learning* (FKL). FKL induces a distribution over kernels by modeling a spectral density (Section 3.1) with a transformed Gaussian process (Section 3.2). Initially we consider one dimensional inputs $x$ and outputs $y$, and then generalize the approach to multiple input dimensions (Section 3.3), and multiple output dimensions (multi-task) (Section 3.4). We consider inference within this model in Section 4.

### 3.1 Spectral Transformations of Kernel Functions

Bochner's Theorem [5, 25] specifies that $k(\cdot)$ is the covariance of a stationary process on $\mathbb{R}$ if and only if

$$k(\tau) = \int_{\mathbb{R}} e^{2\pi i \omega \tau} S(\omega) d\omega, \tag{1}$$

where $\tau = |x - x'|$ is the difference between any pair of inputs $x$ and $x'$, for a positive, finite *spectral density* $S(\omega)$. This relationship is reversible: if $S(\omega)$ is known, $k(\tau)$ can be computed via inverse Fourier transformation.

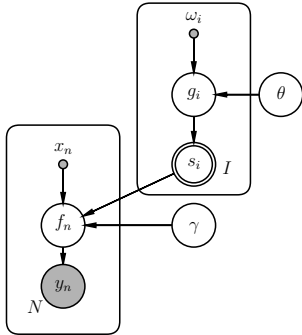

Figure 2: Graphical model for the FKL framework. Observed data is $y_n$, corresponding to the GP output $f_n$. The transformed latent GP is denoted with outputs $S_i$ for observed frequencies $\omega_i$. Hyperparameters are denoted by $\phi = \{\theta, \gamma\}$.

For $k(\tau)$ to be real-valued, $S(\omega)$ must be symmetric. Furthermore, for finitely sampled $\tau$ we are only able to identify angular frequencies up to $2\pi/\Delta$ where $\Delta$ is the minimum absolute difference between any two inputs. Equation 1 simplifies to

$$k(\tau) = \int_{[0, 2\pi/\Delta)} \cos(2\pi\tau\omega) S(\omega) d\omega, \tag{2}$$

by expanding the complex exponential and using the oddness of sine (see Eqs. 4.7 and 4.8 in Rasmussen and Williams [25]) and then truncating the integral to the point of identifiability.

For an arbitrary function, $S(\omega)$, Fourier inversion does not produce an analytic form for $k(\tau)$, however we can use simple numerical integration schemes like the trapezoid rule to approximate the integral in Equation 2 as

$$k(\tau) \approx \frac{\Delta_\omega}{2} \sum_{i=1}^{I} \cos(2\pi\tau\omega_i) S(\omega_i) + \cos(2\pi\tau\omega_{i-1}) S(\omega_{i-1}), \tag{3}$$

where the spectrum is sampled at $I$ evenly spaced frequencies $\omega_i$ that are $\Delta_\omega$ units apart in the frequency domain.

The covariance $k(\tau)$ in Equation (3) is periodic. In practice, frequencies can be chosen such that the period is beyond the bounds that would need to be evaluated in $\tau$. As a simple heuristic we choose $P$ to be $8\tau_{max}$, where $\tau_{max}$ is the maximum distance between training inputs. We then choose frequencies so that $\omega_n = 2\pi n/P$ to ensure $k(\tau)$ is $P$-periodic. We have found choosing 100 frequencies ($n = 0, \ldots, 99$) in this way leads to good performance over a range of experiments in Section 5.

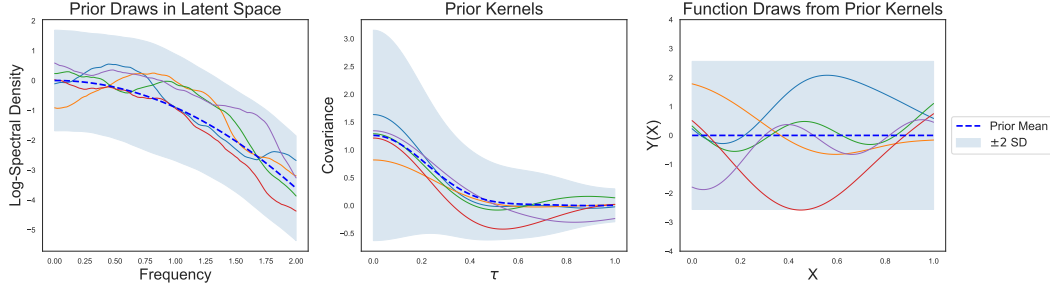

Figure 3: Forward sampling from the hierarchical FKL model of Equation (4). **Left**: Using randomly initialized hyper-parameters $\phi$, we draw functions $g(\omega)$ from the latent GP modeling the log spectral density. **Center**: We use the latent realizations of $g(\omega)$ with Bochner's Theorem and Eq. (3) to compose kernels. **Right**: We sample from a mean-zero Gaussian process with a kernel given by each of the kernel samples. Shaded regions show 2 standard deviations above and below the mean in dashed blue. Notice that the shapes of the prior kernel samples have significant variation but are clearly influenced by the prior mean, providing a controllable inductive bias.

## 3.2 Specification of Latent Density Model

Uniqueness of the relationship in Equation 1 is guaranteed by the Wiener-Khintchine Theorem (see Eq. 4.6 of Rasmussen and Williams [25]), thus learning the spectral density of a kernel is sufficient to learn the kernel. We propose modeling the log-spectral density of kernels using GPs. The log-transformation ensures that the spectral representation is non-negative. We let $\phi = \{\theta, \gamma\}$ be the set of *all* hyper-parameters (including those in both the data, $\gamma$, and latent spaces, $\theta$), to simplify the notation of Section 4.

Using Equation 3 to produce a kernel $k(\tau)$ through $S(\omega)$, the hierarchical model over the data is

$$
\begin{aligned}
\{\text{Hyperprior}\} \qquad & p(\phi) = p(\theta, \gamma) \\
\{\text{Latent GP}\} \qquad & g(\omega)|\theta \sim \mathcal{GP}\left(\mu(\omega; \theta), k_g(\omega, \omega'; \theta)\right) \\
\{\text{Spectral Density}\} \qquad & S(\omega) = \exp\{g(\omega)\} \\
\{\text{Data GP}\} \qquad & f(x_n)|S(\omega), \gamma \sim \mathcal{GP}(\gamma_0, k(\tau; S(\omega))).
\end{aligned}
\tag{4}
$$

We let $f(x)$ be a noise free function that forms part of an observation model. For regression, we can let $y(x) = f(x) + \epsilon(x)$, $\epsilon \sim \mathcal{N}(0, \alpha^2)$ (in future equations we implicitly condition on hyper-parameters of the noise model, e.g., $\alpha^2$, for succinctness, but learn these as part of $\phi$). The approach can easily be adapted to classification through a different observation model; e.g., $p(y(x)) = \sigma(y(x)f(x))$ for binary classification with labels $y \in \{-1, 1\}$. Full hyper-parameter prior specification is given in Appendix 2. Note that unlike logistic Gaussian process density estimators [1, 32] we need not worry about the normalization factor of $S(\omega)$, since it is absorbed by the scale of the kernel over data, $k(0)$. The hierarchical model in Equation 4 defines the functional kernel learning (FKL) prior, with corresponding graphical model in Figure 2. Figure 3 displays the hierarchical model, showing the connection between spectral and data spaces.

A compelling feature of FKL is the ability to conveniently specify a prior expectation for the kernel by specifying a mean function for $g(\omega)$, and to encode smoothness assumptions by the choice of covariance function. For example, if we choose the mean of the latent process $g(\omega)$ to be negative quadratic, then prior kernels are concentrated around RBF kernels, encoding the inductive bias that function values close in input space are likely to have high covariance. In many cases the spectral density contains sharp peaks around dominant frequencies, so we choose a Matérn $3/2$ kernel for the covariance of $g(\omega)$ to capture this behaviour.

## 3.3 Multiple Input Dimensions

We extend FKL to multiple input dimensions by either corresponding each one-dimensional kernel in a product of kernels with its own latent GP with distinct hyper-parameters (FKL separate) or having all one-dimensional kernels be draws from a single latent process with one set of hyper-parameters (FKL shared). The hierarchical Bayesian model over the $d$ dimensions is described in the following manner:

$$\begin{aligned}
\{\text{Hyperprior}\} \qquad & p(\phi) = p(\theta, \gamma) \\
\{\text{Latent GP } \forall d \in \{1, ...D\}\} \qquad & g_d(\omega_d)|\theta \sim \mathcal{GP}\left(\mu(\omega_d; \theta), k_{g_d}(\omega_d, \omega_d'; \theta)\right) \\
\{\text{Product Kernel GP}\} \quad & f(x)|\{g_d(\omega_d)\}_{d=1}^D, \gamma \sim \mathcal{GP}(\gamma_0, \prod_{d=1}^D k(\tau_d; S(\omega_d)))
\end{aligned} \qquad (5)$$

Tying the kernels over each dimension while considering their spectral densities to be draws from the same latent process (FKL shared) provides multiple benefits. Under these assumptions, we have more information to learn the underlying latent GP $g(\omega)$. We also have the helpful inductive bias that the covariance functions across each dimension have some shared high-order properties, and enables linear time scaling with dimensionality.

### 3.4 Multiple Output Dimensions

FKL additionally provides a natural way to view multi-task GPs. We assume that each task (or output), indexed by $t \in \{1, \ldots, T\}$, is generated by a GP with a distinct kernel. The kernels are tied together by assuming each of those $T$ kernels are constructed from realizations of a single *shared* latent GP. Notationally, we let $g(\omega)$ denote the latent GP, and use subscripts $g_t(\omega)$ to indicate independent realizations of this latent GP. The hierarchical model can then be described in the following manner:

$$\begin{aligned}
\{\text{Hyperprior}\} \qquad & p(\phi) = p(\theta, \gamma) \\
\{\text{Latent GP}\} \qquad & g(\omega)|\theta \sim \mathcal{GP}\left(\mu(\omega; \theta), k_g(\omega, \omega'; \theta)\right) \\
\{\text{Task GP } \forall t \in \{1, ...T\}\} \qquad & f_t(x)|g_t(\omega), \gamma \sim \mathcal{GP}(\gamma_{0,t}, k(\tau; S_t(\omega)))
\end{aligned} \qquad (6)$$

In this setup, rather than having to learn the kernel from a single realization of a process (a single task), we can learn the kernel from multiple realizations, which provides a wealth of information for kernel learning [37]. While sharing individual hyper-parameters across multiple tasks is standard (see e.g. Section 9.2 of MacKay [18]), these approaches can only learn limited structure. The information provided by multiple tasks is distinctly amenable to FKL, which shares a flexible *process over kernels* across tasks. FKL can use this information to discover unconventional structure in data, while retaining computational efficiency (see Appendix 1).

## 4 Inference and Prediction

When considering the hierarchical model defined in Equation 4, one needs to learn both the hyper-parameters, $\phi$, and an instance of the latent Gaussian process, $g(\omega)$. We employ alternating updates in which the hyper-parameters $\phi$ and draws of the latent GP are updated separately. A full description of the method is in Algorithm 1 in Appendix 2.

**Updating Hyper-Parameters:** Considering the model specification in Eq. 4, we can define a loss as a function of $\phi = \{\theta, \gamma\}$ for an observation of the density, $\tilde{g}(\omega)$, and data observations $y(x)$. This loss corresponds to the entropy, marginal log-likelihood of the latent GP with fixed data GP, and the marginal log-likelihood of the data GP.

$$\mathcal{L}(\phi) = -\left(\log p(\phi) + \log p(\tilde{g}(\omega)|\theta, \omega) + \log p(y(x)|\tilde{g}(\omega), \gamma, x)\right). \qquad (7)$$

This objective can be optimized using any procedure; we use the AMSGRAD variant of Adam as implemented in PyTorch [26]. For GPs with $D$ input dimensions (and similarly for $D$ output dimensions), we extend Eq. 7 as

$$\mathcal{L}(\phi) = -\left(\log p(\phi) + \sum_{d=1}^D [\log p(\tilde{g}_d(\omega_d)|\theta, \omega)] + \log p(y(x)|\{\tilde{g}_d(\omega_d)\}_{d=1}^D, \gamma, x)\right). \qquad (8)$$

**Updating Latent Gaussian Process:** With fixed hyper-parameters $\phi$, the posterior of the latent GP is

$$p(g(\omega)|\phi, x, y(x), f(x)) \propto \mathcal{N}(\mu(\omega; \theta), k_g(\omega; \theta))p(f(x)|g(\omega), \gamma). \qquad (9)$$

We sample from this posterior using elliptical slice sampling (ESS) [21, 20], which is specifically designed to sample from posteriors with highly correlated Gaussian priors. Note that we must

reparametrize the prior by removing the mean before using ESS; we then consider it part of the likelihood afterwards.

Taken together, these two updates can be viewed as a single sample Monte Carlo expectation maximization (EM) algorithm [33] where only the final $g(\omega)$ sample is used in the Monte Carlo expectation. Using the alternating updates (following Algorithm 1) and transforming the spectral densities into kernels, samples of predictions on the training and testing data can be taken. We generate posterior estimates of kernels by fixing $\phi$ after updating and drawing samples from the posterior distribution, $p(g(\omega)|f, y, \phi)$, taken from ESS (using $y$ as short for $y(x)$, the training data indexed by inputs $x$).

**Prediction:**  The predictive distribution for any test input $x^*$ is given by

$$p(f^*|x^*, x, y, \phi) = \int p(f^*|x^*, x, y, \phi, k)p(k|x^*, x, y, \phi)dk \tag{10}$$

where we are only conditioning on data $x, y$, and hyper-parameters $\phi$ determined from optimization, by *marginalizing* the whole posterior distribution over kernels $k$ given by FKL. We use simple Monte Carlo to approximate this integral as

$$p(f^*|x^*, x, y, \phi) \approx \frac{1}{J}\sum_{j=1}^{J} p(f^*|x^*, x, y, \phi, k_j), \quad k_j \sim p(k|x^*, x, y, \phi). \tag{11}$$

We sample from the posterior over $g(\omega)$ using elliptical slice sampling as above. We then transform these samples $S(\omega) = \exp\{g(\omega)\}$ to form posterior samples from the spectral density. We then sample $k_j \sim p(k|x^*, x, y, \phi)$ by evaluating the trapezoidal approximation in Eq. (3) (at a collection of frequencies $\omega$) for each sample of the spectral density. For regression with Gaussian noise $p(f^*|x^*, x, y, \phi, k)$ is Gaussian, and our expression for the predictive distribution becomes

$$
\begin{aligned}
p(f^*|x^*, x, y, \phi, \omega) &= \frac{1}{J}\sum_{j=1}^{J} \mathcal{N}(\bar{f}^*(x^*)_j, \text{Cov}(f^*)_j) \\
\bar{f}^*(x^*)_j &= k_{f_j}(x^*, x; \gamma)k_{f_j}(x, x; \theta)^{-1}y \\
\text{Cov}(f^*)_j &= k_{f_j}(x^*, x^*; \gamma) - k_{f_j}(x^*, x; \gamma)k_{f_j}(x, x; \theta)^{-1}k_{f_j}(x, x^*; \gamma),
\end{aligned}
\tag{12}
$$

where $k_{f_j}$ is the kernel associated with sample $g_j$ from the posterior over $g$ after transformation to a spectral density and then evaluation of the trapezoidal approximation (suppressing dependence on $\omega$ used in Eq. (3)). $y$ is an $n \times 1$ vector of training data. $k_{f_j}(x, x; \theta)$ is an $n \times n$ matrix formed by evaluating $k_{f_j}$ at all pairs of $n$ training inputs $x$. Similarly $k_{f_j}(x^*, x^*; \theta)$ is a scalar and $k_{f_j}(x^*, x)$ is $1 \times n$ for a single test input $x^*$. This distribution is a mixture of Gaussians with $J$ components. Following the above procedure, we obtain $J$ samples from the unconditional distribution in Eq. (12). We can compute the sample mean for point predictions and twice the sample standard deviation for a credible set. We use the mixture of Gaussians representation in conjunction with the laws of total mean and variance to approximate the moments of the predictive distribution in Eq. (12), which is what we do for the experiments.

## 5  Experiments

We demonstrate the practicality of FKL over a wide range of experiments: (1) recovering known kernels from data (Section 5.1); (2) extrapolation (Section 5.2); (3) multi-dimensional inputs and irregularly spaced data (section 5.3); (4) multi-task precipitation data (Section 5.4); and (5) multidimensional pattern extrapolation (Section 5.5). We compare to the standard RBF and Matérn kernels, as well as spectral mixture kernels [34], and the Bayesian nonparametric spectral estimation (BNSE) of Tobar [30].

For FKL experiments, we use $g(\omega)$ with a negative quadratic mean function (to induce an RBF-like prior mean in the distribution over kernels), and a Matérn kernel with $\nu = \frac{3}{2}$ (to capture the typical sharpness of spectral densities). We use the heuristic for frequencies in the trapezoid rule described in Section 3.1. Using $J = 10$ samples from the posterior over kernels, we evaluate the sample mean and twice the sample standard deviation from the unconditional predictive distribution in Eq. (12) for point predictions and credible sets. We perform all experiments in GPyTorch [10].

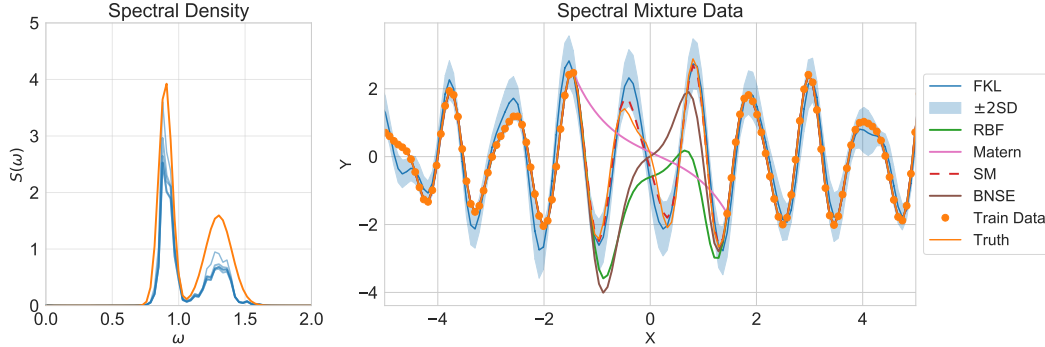

Figure 4: **Left**: Samples from the FKL posterior over the spectral density capture the shape of the true spectrum. **Right**: Many of the FKL predictions on the held out data are nearly on par with the ground-truth model (SM in dashed red). GPs using the other kernels perform poorly on extrapolation away from the training points.

## 5.1 Recovery of Spectral Mixture Kernels

Here we test the ability of FKL to recover known ground truth kernels. We generate 150 data points, $x_i \sim U(-7., 7)$ randomly and then draw a random function from a GP with a two component spectral mixture kernel with weights 1 and 0.5, spectral means of 0.2 and 0.9 and standard deviations of 0.05. As shown in Figure 4, FKL accurately reconstructs the underlying spectral density, which enables accurate in-filling of data in a held out test region, alongside reliable credible sets. A GP with a spectral mixture kernel is suited for this task and closely matches with withheld data. GP regression with the RBF or Matérn kernels is unable to predict accurately very far from the training points. BNSE similarly interpolates the training data well but performs poorly on the extrapolation region away from the data. In Appendix 5.1 we illustrate an additional kernel recovery experiment, with similar results.

## 5.2 Interpolation and Extrapolation

**Airline Passenger Data**    We next consider the airline passenger dataset [14] consisting of 96 monthly observations of numbers of airline passengers from 1949 to 1961, and attempt to extrapolate the next 48 observations. We standardize the dataset to have zero mean and unit standard deviation before modeling. The dataset is difficult for Gaussian processes with standard stationary kernels, due to the rising trend, and difficulty in extrapolating quasi-periodic structure.

**Sinc**    We model a pattern of three sinc functions replicating the experiment of Wilson and Adams [34]. Here $y(x) = \text{sinc}(x + 10) + \text{sinc}(x) + \text{sinc}(x - 10)$ with $\text{sinc}(x) = \sin(\pi x)/(\pi x)$. This has been shown previously [34] to be a case for which parametric kernels fail to pick up on the correct periodic structure of the data.

Figures 5a and 5b show that FKL outperforms simple parametric kernels on complex datasets. Performance of FKL is on par with that of SM kernels while requiring less manual tuning and being more robust to initialization.

## 5.3 Multiple Dimensions: Interpolation on UCI datasets

We use the product kernel described in Section 5.3 with both separate and shared latent GPs for regression tasks on UCI datasets. Figure 6 visually depicts the model with respect to prior and posterior products of kernels. We standardize the data to zero mean and unit variance and randomly split the training and test sets, corresponding to 90% and 10% of the full data, respectively. We conduct experiments over 10 random splits and show the average RMSE and standard deviation. We compare to the RBF, ARD, and ARD Matérn. Furthermore, we compare the results of sharing a single latent GP across the kernels of the product decomposition(Eq. 5) with independent latent GPs for each kernel in the decomposition.

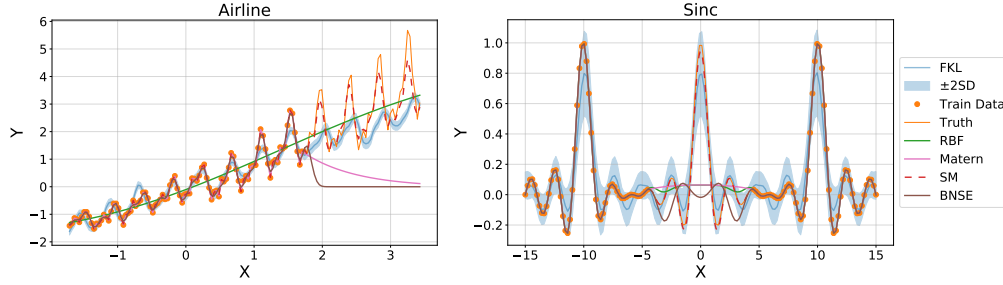

(a) Extrapolation on the airlines dataset [14].　　　(b) Interpolation on the sinc function.

Figure 5: **(a)**: Extrapolation on the airline passenger dataset. **(b)**: Prediction on sinc data. FKL is on par with a carefully tuned SM kernel (dashed pink) in **(a)** and shows best performance in **(b)**, BNSE (brown) performs well on the training data, but quickly reverts to the mean in the testing set.

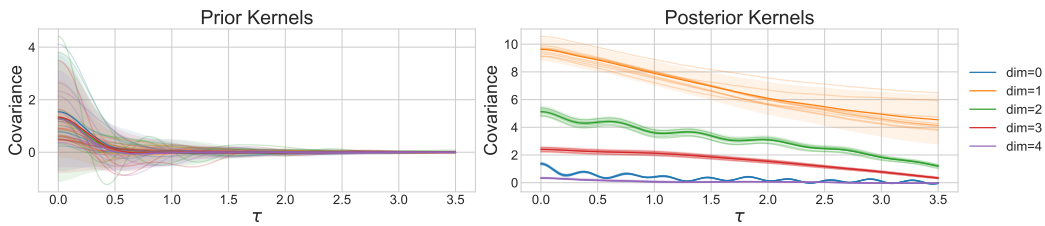

Figure 6: Samples of prior **(a)** and posterior **(b)** kernels displayed alongside the sample mean (thick lines) and $\pm$ 2 standard deviations (shade). Each color corresponds to a kernel, $k(\cdot)$, for a dimension of the airfoil dataset.

## 5.4　Multi-Task Extrapolation

We use the multi-task version of FKL in Section 3.4 to model precipitation data sourced from the United States Historical Climatology Network [19]. The data contain daily precipitation measurements over 115 years collected at 1218 locations in the US. Average positive precipitation by day of the year is taken for three climatologically similar recording locations in Colorado: Boulder, Telluride, and Steamboat Springs, as shown in Figure 8. The data for these locations have similar seasonal variations, motivating a shared latent GP across tasks, with a flexible kernel process capable of learning this structure. Following the procedure outlined in Section 4 and detailed in Algorithm 2 in the Appendix, FKL provides predictive distributions that accurately interpolates and extrapolates the data with appropriate credible sets. In Appendix 6 we extend these multi-task precipitation results to large scale experimentation with datasets containing tens of thousands of points.

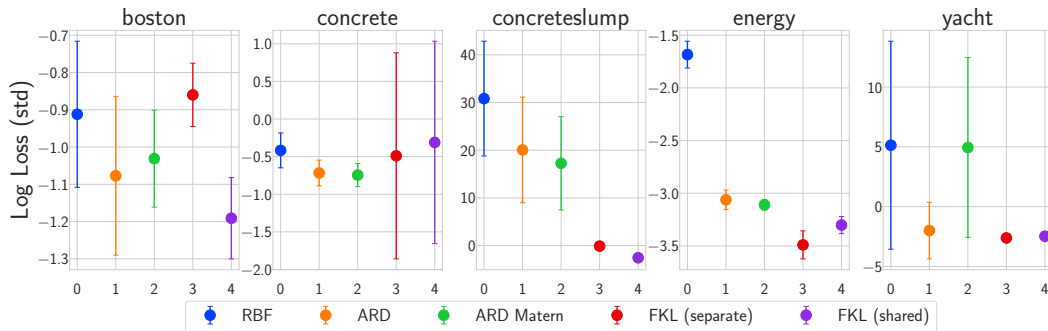

Figure 7: Standardized log losses on five of the 12 UCI datasets used. Here, we can see that FKL typically outperforms parametric kernels, even with a shared latent GP. See Table 2 for the full results in the Appendix.

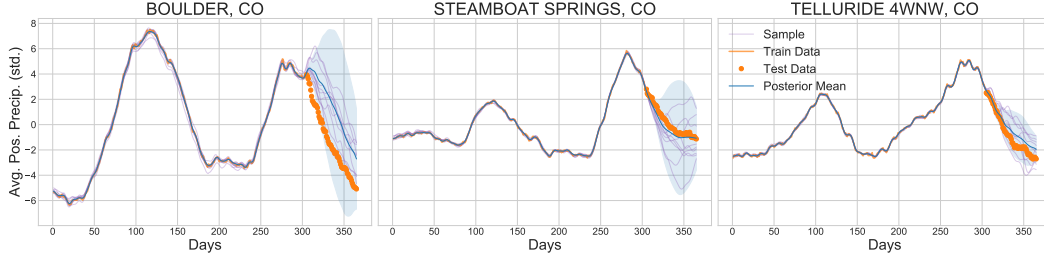

Figure 8: Posterior predictions generated using latent GP samples. 10 samples of the latent GP for each site are used to construct covariance matrices and posterior predictions of the GPs over the data.

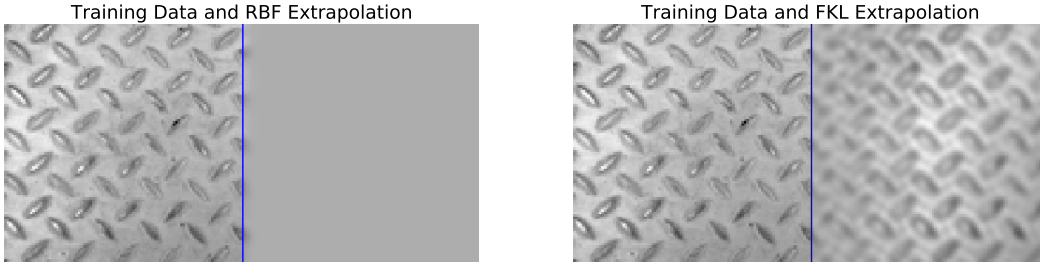

Figure 9: Texture Extrapolation: training data is shown to the left of the blue line and predicted extrapolations according to each model are to the right.

## 5.5 Scalability and Texture Extrapolation

Large datasets typically provide additional information to learn rich covariance structure. Following the setup in [36], we exploit the underlying structure in images and scale FKL to learn such a rich covariance — enabling extrapolation on textures. When the inputs, $X$, form a Cartesian product multidimensional grid, the covariance matrix decomposes as the Kronecker product of the covariance matrices over each input dimension, i.e. $K(X, X) = K(X_1, X_1) \otimes K(X_2, X_2) \otimes \cdots \otimes K(X_P, X_P)$ where $X_i$ are the elements of the grid in the $i^{th}$ dimension [28]. Using the eigendecompositions of Kronecker matrices, solutions to linear systems and log determinants of covariance matrices that have Kronecker structure can be computed exactly in $\mathcal{O}(PN^{P/2})$ time, instead of the standard cubic scaling in $N$ [36].

We train FKL on a $10,000$ pixel image of a steel tread-plate and extrapolate the pattern beyond the training domain. As shown in Figure 9, FKL uncovers the underlying structure, with no sophisticated initialization procedure. While the spectral mixture kernel performs well on these tasks [36], it requires involved initialization procedures. By contrast, standard kernels, such as the RBF kernel, are unable to discover the covariance structure to extrapolate on these tasks.

## 6 Discussion

In an era where the number of model parameters often exceeds the number of available data points, the function-space view provides a more natural representation of our models. It is the complexity and inductive biases of *functions* that affect generalization performance, rather than the number of parameters in a model. Moreover, we can interpretably encode our assumptions over functions, whereas parameters are often inscrutable. We have shown the function-space approach to learning covariance structure is flexible and convenient, able to automatically discover rich representations of data, without over-fitting.

There are many exciting directions for future work: (i) interpreting the learned covariance structure across multiple heterogeneous tasks to gain new scientific insights; (ii) developing function-space distributions over *non-stationary* kernels; and (iii) developing deep hierarchical functional kernel learning models, where we consider function space *distributions over distributions* of kernels.

## Acknowledgements

GWB, WJM, JPS, and AGW were supported by an Amazon Research Award, Facebook Research, NSF IIS-1563887, and NSF IIS-1910266. WJM was additionally supported by an NSF Graduate Research Fellowship under Grant No. DGE-1650441.

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
