[Supplementary Material]

# Supplementary Materials for Function-Space Distributions over Kernels

**Gregory W. Benton**[*1]   **Wesley J. Maddox**[*2]   **Jayson P. Salkey**[*1]
**Júlio Albinati**[3]   **Andrew Gordon Wilson**[1,2]

[1]Courant Institute of Mathematical Sciences, New York University
[2]Center for Data Science, New York University
[3]Microsoft

## 1   Computational Complexity

Note that when sampling at $N$ data points and $I$ frequencies, the storage costs for this model are naively $\mathcal{O}(N^2 + I^2)$ with the computational cost for prediction of $\mathcal{O}(N^3 + I^3)$. Using pre-conditioned conjugate gradients for inverses and stochastic Lanczos quadrature (SLQ) or the log determinants [7] as implemented in GPyTorch [8] for the data and likelihood calls can immediately reduce the computational cost to $\mathcal{O}(N^2 + I^2)$. However, the randomness in the log determinant calculations proved to be problematic for ESS and we only used SLQ for the gradient-based updates, keeping the overall time complexity cubic in $N$. Given that the latent Gaussian processes are on a pre-defined grid, we can utilize fast Toeplitz matrix multiplications [15] to reduce the time complexity to $\mathcal{O}(N^3 + I \log I)$ and the memory complexity to $\mathcal{O}(N^3 + I)$.

Extending the model to multi-dimensional inputs and multiple outputs adds on a linear term for both dimensionality $D$ and tasks $T$ independently, so for a multi-task model with $T$ tasks predictions are done in $\mathcal{O}(T(N^3 + I))$. Note that this is significant improvement over the $\mathcal{O}(T^3 N^3)$ needed to do exact inference in previous multi-task work such as Bonilla et al. [4].

For enhanced scalability, we can approximate the kernel matrices in single (and low) dimensions by utilizing scalable kernel interpolation (SKI) as introduced by Wilson and Nickisch [14]. Using $m$ inducing points we can achieve an inference cost of $\mathcal{O}(N + m \log m + I \log I)$ or $\mathcal{O}(T(N + m \log m + I \log I))$ for the multi-task setting.

## 2   Latent Model Specification

### 2.1   Initialization

FKL proves to be robust to initialization, thus for simplicity we initialize the spectral density to be constant, $S(\omega) = 1$, for a large range of frequencies. An experiment detailing the models robustness is given in the Appendix.

### 2.2   Specification of the Latent GP

We fix the mean and covariance of the latent process $g(\omega)$ to take the following forms:

$$\{\text{log of RBF spectral density}\} \qquad \mu(\omega;\theta) = \theta_0 - \frac{\omega^2}{2\tilde{\theta_1}^2}$$

$$\{\text{Matérn kernel}\} \qquad k_g(\omega,\omega';\theta) = \frac{2^{1-\nu}}{\Gamma(\nu)} \left(\sqrt{2\nu}\frac{|\omega - \omega'|}{\tilde{\theta}_2}\right) K_\nu \left(\sqrt{2\nu}\frac{|\omega - \omega'|}{\tilde{\theta}_2}\right) + \tilde{\theta}_3 \delta_{\tau=0} \tag{1}$$

---

[*]Equal contribution.

---

**Algorithm 1** Alternating Sampler

---

**Input:** Data $(x, y)$, Initial hyper-parameters $\phi_0$, Sampling frequencies $\omega$, Initial Latent GP $g(\omega)$, Number of gradient steps to take per iteration $N_{optim}$, Number of ESS samples per update per iteration $N_{ESS}$,
**repeat**
   **for** $i = 1$ **to** $N_{optim}$ **do**
      Update $\phi$ using gradient descent given $g(\omega)$ and Eqn. 7
   **end for**
   **for** $i = 1$ **to** $N_{ESS}$ **do**
      Update $g(\omega)$ using elliptical slice sampling given $\phi$ and Eqn. 9
   **end for**
**until** convergence

---

---

**Algorithm 2** Multi-Task Alternating Sampler

---

**Input:** Data $(x, Y)$, Initial hyper-parameters $\phi_0$, Sampling frequencies $\omega$, Initial Latent GPs $g_i(\omega)$ for $i = 1, \ldots, T$, Number of gradient steps to take per iteration $N_{optim}$, Number of ESS samples per update per iteration $N_{ESS}$,
**repeat**
   **for** $i = 1$ **to** $N_{optim}$ **do**
      Update $\phi$ using gradient descent given $g(\omega)$ and Eqn. 8
   **end for**
   **for** $t = 1$ **to** $T$ **do**
      **for** $i = 1$ **to** $N_{ESS}$ **do**
         Update $g_t(\omega)$ using elliptical slice sampling given $\phi$ and Eqn. 9 with respect to $f_t(x)$
      **end for**
   **end for**
**until** convergence

---

The $\tilde{\theta}_i$'s are non-negative variables, so are computed with $\tilde{\theta}_i = \log(e^{\theta_i} + 1)$, the softplus of the raw value. The mean parametrization coupled with the constraints fixes the latent mean to be negative quadratic, like the logarithm of an RBF spectral density.

### 2.3   Prior Specification

For the noise terms, we place smoothed box priors[2] on the range (1e-8, 1e-3) to control both numerical instability and the noise terms. For the constant mean terms in both the data and latent means, we place uninformative $\mathcal{N}(0, 100)$ priors. For the length-scale in the spectral density mean along with the length-scale and output-scale of the covariance of the spectral density GP, we place standard log-normal priors.

## 3   Density and Error Bounds of FKL

### 3.1   Error Rate of Trapezoidal Rule Approximation

Given a sample path from a Gaussian process with a Matérn kernel as is used in our implementation, we can get explicit $O(1/I)$ error bounds on the error of trapezoid rule integration of the warped GP instead by checking Holder continuity of sample draws from the latent GP [3], and using results on the error of trapezoid rule for Holder continuous functions [5]. Note that we could use standard error bounds if we use a GP with twice differentiable sample paths.

## 3.2 Density Amongst Stationary Kernels

We next note that the trapezoidal rule is just a finite sample version of both Riemann and Darboux integrals. Thus, functional kernel learning can also be written as a linear combination of the trigonemetric basis expansions and the spectral density (e.g. in sparse spectrum form like Lázaro-Gredilla et al. [9]). Thus, FKL can model discontinuous but finite measures because mixtures of Gaussians are dense approximations of Riemann integrable densities (see Theorem 5 of Shen et al. [12]). Thus, the trapezoid rule will be an approximator of the true kernel on the compact set $[0, \omega_{max}]$, converging as $\omega_{max} \to \infty$ (e.g. as the number of basis functions goes to infinity).

Finally, we note that in the multi-dimensional case, FKL does not provide support over all stationary covariances (like other spectral approaches [12, 13]), but we find in practice that the domain of support is great enough for accurate performance on most tasks. We would need to at least model the $\omega$'s for each dimension on a grid to provide full support, at a cost of af the number of grid points exponentially increasing. Future work will help to alleviate this issue.

# 4 Sensitivity to Initialization

Part of the strength of FKL, particularly over competing methods like spectral mixture (SM) kernels, is robustness to initialization. We compare the performance of FKL and SM kernels on interpolating data generated from a GP with a quasi-periodic kernel.

In GPyTorch spectral mixture (SM) kernels are initialized to,

$$\mu = \log(\exp(0) + 1)$$
$$\sigma = \log(\exp(0) + 1)$$
$$w = \log(\exp(0) + 1),$$

i.e. the means, variances, and weights of each mixture component is the softplus of 0 prior to calling the data initialization routine [8]. The data-based initialization routine uses statistics of the data to randomly initialize the parameters of the mixture components, and performance is highly dependent on this initialization.

In the current implementation FKL is initialized with a spectral density that is constant,

$$S_0(\omega) = 1 \quad \forall \omega$$
$$g_0(\omega) = 0 \quad \forall \omega,$$

where $g(\omega)$ is the log-spectral density, which is modeled using a latent GP. The surprising fact, and what makes FKL such an appealing model for complex problems, is robustness to initialization. In practice we see no gains in predictive performance when initializing in a more sophisticated fashion than is currently done. This robustness goes far enough that we don't even see performance gains when we have access to ground truth data and can initialize the spectral density to be near to the spectral density of the kernel of generative model itself.

Data are generated using a GP with quasi-periodic kernel and the middle portion of the data are held out as a testing set. Using the inverse Fourier transform we can compute the spectral density of the generating quasi-periodic kernel directly, $S^*(\omega)$. First we train and predict using a SM kernel that is has parameters initialized to the constant values from above, and compare to a SM kernel using GPyTorch's built in data-based initialization. Next we repeat the procedure using a default initialized FKL model, then compare to an FKL model where the spectral density has been initialized to a corrupted version of the ground truth spectral density. Thus we compare FKL models with the initializations,

$$S_0(\omega) = 1 \quad \forall \omega$$
$$S_0(\omega) = S^*(\omega) + \mathcal{N}(0, 0.1) \quad \forall \omega.$$

The results are shown in Figures 1 and 2. What we see is that a naive implementation of SM kernels leads to poor performance on the testing set, while FKL performs nearly the same whether we initialize the spectral density to an arbitrary value, or to nearly the ground truth.

Figure 1: Comparison of naive and data-based initialized SM kernels on interpolation tasks. **Left**: the default (naive) initialized kernel, **Right**: the data-based initialized kernel.

Figure 2: Comparison of basic and ground-truth initialized FKL kernels on interpolation tasks. **Left**: the default (naive) initialized kernel, **Right**: the ground-truth initialized kernel.

## 5 Further Experiments

### 5.1 Recovery of Known Kernels

**Spectral Mixture Kernel**  Extending from Section 5.1, we also display the accuracy of the kernel reconstruction given the samples drawn in the latent space. Figure 3 shows the accurately sampled spectral density, and the kernels reconstructed from these samples.

**Quasi-Periodic Kernel**  Synthetic data are generated from a mean zero Gaussian process with kernel,

$$k(\tau; \ell, \omega) = \exp\left(-\frac{\tau^2}{2\ell^2}\right) \exp\left(-2\sin^2(\pi\tau\omega)\right). \tag{2}$$

Since there is inherent periodicity in the generative model, the true spectral density has distinct modes corresponding to the period length of the sinusoidal component of the kernel. The spectral density of this kernel is not analytically computed, however using the known kernel the discrete Fourier transform allows an approximation of the ground-truth spectrum to be found, and comparison in the spectral domain can be made.

Using this latent GP model accurate reconstruction of both the spectral density and kernel are obtained using only training data. Further more, infilling into the testing set shows high accuracy and the confidence region encompasses the data.

### 5.2 Foreign Exchange Rates Dataset

We consider multi-output prediction tasks on a foreign exchange rates dataset originally developed in [1]. The dataset consists of the exchange rates of 10 currencies and 3 precious metals with respect to

Figure 3: Samples from the latent GP displayed in the spectral domain along with the ground truth (Left) and the reconstructed kernels generated by these samples (Right).

Table 1: Standardized mean squared error on FX dataset. Comparisons are with independent Gaussian processes (IGP), convolved multi-output GP (CMOGP) [2], collaborative GP (CGP) [10], and Gaussian process autoregressive model (GPAR). Note that the GPAR is perhaps best viewed as a deep Gaussian process with known inputs. Comparisons taken from [11]. Note that FKL multi-task outperforms the standard multi-task GP methods) averaged over 10 random trials.

| Model | IGP | CMOGP | CGP | GPAR | FKL(multi-task) |
|---|---|---|---|---|---|
| SMSE | 0.5996 | 0.2427 | 0.2125 | **0.0302** | $0.1392 \pm 0.01$ |

the US dollar in 2007. The task is to predict the Canadian dollar (CAD) on days 50-100, Japanese yen (JPY) on days 100-150, and Australian dollar (AUD) on days 150-200, given the exchange rate information for all other days. Due to market differences, there are occasionally also missing data. Like in [11], we measure performance with the standardized mean square error (SMSE). The results from this experiment are shown in Table 1 with comparisons taken from both [11] and [10]. FKL performs considerably better than both types of collaborative Gaussian process, which constrain the outputs considerably more. By comparison, the GPAR [11] outperforms FKL on this task, perhaps due to its explicit ordering of tasks and its increased depth (the GPAR is a special case of deep Gaussian processes [6]).

Here, we utilize 5 rounds of the alternating sampler with 10 optimization and 50 ESS iterations and run on a single GPU (with 10 repetitions taking about 3 minutes).

### 5.3 UCI Tables

Tables 2, 3, and 4 show the RMSE, standardized log loss, and negative log likelihoods of FKL (both separate and shared latent models) compared to standard parametric models on UCI regression tasks.

## 6 Large-Scale Precipitation Extrapolation

We demonstrate the scalability and practicality of FKL by extending this to a much larger dataset; modeling 108 different stations in seven American states across the northeast (ME, MA, VT, NH, RI, CT, NY) with a single latent Gaussian process, training on the first 300 days of the year, and attempting to extrapolate on the final 65 days. Despite not including any geographic information (e.g. longitude and latitude), FKL fits the trends across this climatologically diverse region We show extrapolation on 120 stations in Figure 15 in the Appendix. Note that this corresponds to a dataset size of greater than 30,000 data points, and that we were able to fit this dataset on a single Nvidia 1080 Ti GPU in roughly 30 minutes.

Figure 4: Spectrum (Above Left) and kernel (Above Right) reconstruction, and resulting data prediction (Below) for data generated by a quasi-periodic kernel.

Table 2: UCI Regression RMSEs, comparisons are with RBF, ARD, and ARD Matérn kernels, $N$ points $D$ input dimensions. We compare to separate latent GPs for each input dimension, finding that sharing a single latent GP across dimensions works better than both the standard fixed spectrum approaches and separate latent GPs. Each of the experiments were conducted 10 times with random 90/10 train/test splits and we report the average RMSE $\pm$ one standard deviation.

| Dataset | $N$ | $D$ | RBF | ARD | ARD Matérn | FKL-PB (separate) | FKL-PB (shared) |
|---|---|---|---|---|---|---|---|
| challenger | 23 | 4 | $0.713 \pm 0.348$ | $0.659 \pm 0.368$ | $0.612 \pm 0.268$ | $0.58 \pm 0.225$ | $\mathbf{0.548 \pm 0.174}$ |
| fertility | 100 | 9 | $0.159 \pm 0.036$ | $0.177 \pm 0.035$ | $\mathbf{0.148 \pm 0.038}$ | $0.19 \pm 0.047$ | $0.182 \pm 0.022$ |
| concreteslump | 103 | 7 | $36.302 \pm 7.934$ | $27.377 \pm 7.782$ | $26.335 \pm 7.482$ | $59.444 \pm 12.879$ | $\mathbf{4.385 \pm 1.332}$ |
| servo | 167 | 4 | $0.305 \pm 0.056$ | $\mathbf{0.23 \pm 0.075}$ | $0.256 \pm 0.06$ | $0.282 \pm 0.086$ | $0.288 \pm 0.063$ |
| yacht | 308 | 6 | $0.17 \pm 0.07$ | $0.187 \pm 0.078$ | $0.269 \pm 0.048$ | $0.193 \pm 0.13$ | $\mathbf{0.11 \pm 0.054}$ |
| autompg | 392 | 7 | $2.651 \pm 0.488$ | $3.077 \pm 0.544$ | $\mathbf{2.516 \pm 0.332}$ | $2.838 \pm 0.374$ | $2.69 \pm 0.492$ |
| housing | 506 | 13 | $3.771 \pm 0.675$ | $3.222 \pm 0.846$ | $3.261 \pm 0.624$ | $4.679 \pm 0.632$ | $\mathbf{2.703 \pm 0.227}$ |
| stock | 536 | 11 | $\mathbf{0.005 \pm 0.001}$ | $\mathbf{0.005 \pm 0.001}$ | $\mathbf{0.005 \pm 0.001}$ | $0.018 \pm 0.002$ | $0.016 \pm 0.001$ |
| pendulum | 630 | 9 | $1.297 \pm 0.315$ | $1.185 \pm 0.326$ | $\mathbf{1.013 \pm 0.207}$ | $2.747 \pm 0.737$ | $1.562 \pm 0.554$ |
| energy | 768 | 8 | $1.839 \pm 0.253$ | $0.457 \pm 0.035$ | $0.373 \pm 0.062$ | $\mathbf{0.296 \pm 0.066}$ | $0.334 \pm 0.063$ |
| concrete | 1030 | 8 | $7.001 \pm 0.513$ | $6.125 \pm 0.456$ | $6.058 \pm 0.373$ | $\mathbf{3.781 \pm 0.501}$ | $4.047 \pm 0.693$ |
| airfoil | 1503 | 5 | $2.503 \pm 0.202$ | $1.696 \pm 0.243$ | $1.595 \pm 0.296$ | $\mathbf{1.378 \pm 0.176}$ | $1.39 \pm 0.181$ |

Table 3: UCI Regression Mean Standardized Log loss, comparisons are with RBF, ARD, and ARD Matérn kernels, $N$ points $D$ input dimensions. We compare to separate latent GPs for each input dimension. Each of the experiments were conducted 10 times with a random 90/10 train/test split and reported over $\pm$ a standard deviation.

| Dataset | $N$ | $D$ | RBF | ARD | ARD Matérn | FKL-PB (separate) | FKL-PB (shared) |
|---|---|---|---|---|---|---|---|
| challenger | 23 | 4 | $0.83 \pm 1.085$ | $0.91 \pm 1.951$ | $0.383 \pm 0.778$ | $\mathbf{-0.053 \pm 0.192}$ | $0.216 \pm 0.292$ |
| fertility | 100 | 9 | $-0.049 \pm 0.075$ | $\mathbf{-0.094 \pm 0.137}$ | $-0.077 \pm 0.295$ | $0.013 \pm 0.06$ | $-0.0 \pm 0.017$ |
| concreteslump | 103 | 7 | $30.821 \pm 12.039$ | $20.055 \pm 11.079$ | $17.247 \pm 9.789$ | $-0.125 \pm 0.131$ | $\mathbf{-2.57 \pm 0.23}$ |
| servo | 167 | 4 | $-1.076 \pm 0.216$ | $-1.242 \pm 0.386$ | $-1.25 \pm 0.121$ | $\mathbf{-1.28 \pm 0.218}$ | $-0.981 \pm 0.272$ |
| yacht | 308 | 6 | $5.136 \pm 8.696$ | $-2.001 \pm 2.369$ | $4.943 \pm 7.521$ | $\mathbf{-2.62 \pm 0.225}$ | $-2.477 \pm 0.17$ |
| autompg | 392 | 7 | $-1.065 \pm 0.216$ | $-0.93 \pm 0.306$ | $\mathbf{-1.085 \pm 0.152}$ | $-1.034 \pm 0.149$ | $-0.888 \pm 0.482$ |
| boston | 506 | 13 | $-0.912 \pm 0.196$ | $-1.077 \pm 0.213$ | $-1.031 \pm 0.13$ | $-0.86 \pm 0.085$ | $\mathbf{-1.191 \pm 0.109}$ |
| stock | 536 | 11 | $-0.831 \pm 0.082$ | $-0.82 \pm 0.088$ | $\mathbf{-0.868 \pm 0.105}$ | $0.014 \pm 0.04$ | $-0.001 \pm 0.017$ |
| pendulum | 630 | 9 | $-1.12 \pm 0.084$ | $-1.358 \pm 0.147$ | $-1.586 \pm 0.227$ | $-0.323 \pm 0.181$ | $\mathbf{-1.685 \pm 0.263}$ |
| energy | 768 | 8 | $-1.684 \pm 0.127$ | $-3.062 \pm 0.093$ | $-3.11 \pm 0.05$ | $\mathbf{-3.49 \pm 0.133}$ | $-3.302 \pm 0.081$ |
| concrete | 1030 | 8 | $-0.417 \pm 0.232$ | $-0.717 \pm 0.171$ | $\mathbf{-0.745 \pm 0.154}$ | $-0.489 \pm 1.37$ | $-0.311 \pm 1.345$ |
| airfoil | 1503 | 5 | $-0.994 \pm 0.064$ | $-1.177 \pm 0.078$ | $-1.31 \pm 0.048$ | $-1.448 \pm 0.336$ | $\mathbf{-1.586 \pm 0.198}$ |

Table 4: UCI Regression Negative Log-likelihoods, comparisons are with RBF, ARD, and ARD Matérn kernels, $N$ points $D$ input dimensions. We compare to separate latent GPs for each input dimension. Each of the experiments were conducted 10 times with a random 90/10 train/test split and reported over $\pm$ a standard deviation.

| Dataset | $N$ | $D$ | RBF | ARD | ARD Matérn | FKL-PB (separate) | FKL-PB (shared) |
|---|---|---|---|---|---|---|---|
| challenger | 23 | 4 | $5.74 \pm 4.547$ | $6.064 \pm 7.283$ | $3.753 \pm 3.05$ | $\mathbf{2.82 \pm 0.809}$ | $2.966 \pm 0.854$ |
| fertility | 100 | 9 | $-3.901 \pm 1.76$ | $-2.861 \pm 2.187$ | $\mathbf{-4.408 \pm 2.582}$ | $-1.83 \pm 3.336$ | $-2.738 \pm 1.252$ |
| concreteslump | 103 | 7 | $400.451 \pm 134.157$ | $282.544 \pm 124.796$ | $250.299 \pm 108.762$ | $60.248 \pm 2.542$ | $\mathbf{33.016 \pm 1.965}$ |
| servo | 167 | 4 | $5.144 \pm 3.995$ | $1.101 \pm 5.871$ | $1.374 \pm 3.14$ | $\mathbf{0.93 \pm 3.867}$ | $4.686 \pm 5.271$ |
| yacht | 308 | 6 | $221.42 \pm 271.437$ | $1.65 \pm 76.479$ | $\mathbf{-19.949 \pm 14.092}$ | $-15.703 \pm 8.233$ | $-14.52 \pm 4.7$ |
| autompg | 392 | 7 | $96.189 \pm 8.025$ | $104.563 \pm 13.36$ | $\mathbf{94.012 \pm 5.033}$ | $98.942 \pm 6.135$ | $101.757 \pm 19.333$ |
| housing | 506 | 13 | $139.617 \pm 11.546$ | $131.22 \pm 15.034$ | $130.841 \pm 10.506$ | $143.75 \pm 5.714$ | $\mathbf{122.618 \pm 3.91}$ |
| stock | 536 | 11 | $\mathbf{-191.624 \pm 1.626}$ | $-191.515 \pm 1.472$ | $-191.154 \pm 1.318$ | $-140.055 \pm 6.679$ | $-147.805 \pm 2.577$ |
| pendulum | 630 | 9 | $84.964 \pm 3.402$ | $69.371 \pm 7.299$ | $62.64 \pm 5.692$ | $141.121 \pm 20.914$ | $\mathbf{53.86 \pm 16.301}$ |
| energy | 768 | 8 | $157.1 \pm 8.894$ | $52.118 \pm 5.835$ | $47.776 \pm 3.591$ | $\mathbf{17.808 \pm 9.927}$ | $30.222 \pm 6.881$ |
| concrete | 1030 | 8 | $395.596 \pm 21.02$ | $361.792 \pm 20.077$ | $\mathbf{357.248 \pm 14.532}$ | $384.242 \pm 140.779$ | $405.779 \pm 137.561$ |
| airfoil | 1503 | 5 | $358.932 \pm 8.932$ | $325.059 \pm 6.605$ | $305.588 \pm 7.462$ | $284.895 \pm 48.796$ | $\mathbf{270.073 \pm 28.424}$ |

Figure 5: 40 stations modelled in the multi-task extrapolation test. The multi-task FKL both interpolates and extrapolates well even for relatively geographically diverse datasets.

Figure 6: 40 stations modelled in the multi-task extrapolation test. The multi-task FKL both interpolates and extrapolates well even for relatively geographically diverse datasets.

Figure 7: 40 stations modeled in the multi-task extrapolation test. The multi-task FKL both interpolates and extrapolates well even for relatively geographically diverse datasets.

Figure 8: 17 stations modeled in the multi-task extrapolation test. The multi-task FKL both interpolates and extrapolates well even for relatively geographically diverse datasets.

Figure 9: Map of locations used for large scale multi task experiment.

## Footnotes

[2]A smooth approximation to uniform priors, where $B(x) = \{a \leq x \leq b\}$ then $d(x, B) := \min_{x' \in B} |x - x'|$ and finally the density is given by $f(x) := \exp\{-d(x, B)^2/\sqrt{2\sigma^2}\}$. See `https://gpytorch.readthedocs.io/en/latest/priors.html` for further implementation details.