[Reviews · NeurIPS 2019]

Reviewer 1



The authors have given a novel and expressive manner to learn covariance functions for Gaussian processes. The explanation of the model and the inference procedure are clear and to the point, making it a pleasant read. However, I think section 5 can be improved, perhaps swapping in material in the supplementary paper. Main points 1. A discussion on how one might approach non-stationary kernels may help to initiate future work. 2. A discussion on how one might approach non-axis aligned kernels for multi-dimension input would also be helpful. 3. Line 112. If one need not worry about the normalization factor, does it mean that it is redundant to augment this kernel with a signal-variance factor? 4. Eq. 7. Seems to be missing a few subscript t's for tasks. 5. Section 5. Good to re-iterate how $\Delta$ in equation 2 is obtained, esp for multi-dimensional data. 6. Section 5.1 Please refer to Figure 3. 7. Section 5.2 What is the model predicting for the airline passenger data set? Is it the number of passengers (please indicate)? Why are there negative values in Figure 4a? Would modelling the logarithm of the data make more sense? 8. Figure 4. The lines are too close to prove the point. Please use a zoomed inset. 9. Section 5.3. Are there categorical and nominal data in the data set? If so, how useful/sensible is it to use the spectral method? Also a rough investigation into why the FKL does poorly on the fertility data set will be very helpful. 10. Section 5.4. Please discuss the figure (Figure 5) that you actually place on the main text, rather than go on the figures in the supplementary paper. The divide between the main paper and the supplementary paper is not reasonable. Minor points A. References [22] and [23] are duplicated. B. Lines 118 to 122 reads awkward. Suggest to paraphrase C. Lines 195 to 197. Suggest to use "linear combination" rather than "mixture", since the mixture models have the weights sum to unity.

Reviewer 2



Review update: Thanks for the response. It addressed my concerns well, and the SM kernel comparison seemed to give consistent results. I'm increasing my score slightly. --- This paper proposes functional kernel learning (FKL) a Bayesian nonparametric framework to infer flexible, stationary kernels for Gaussian process (GP) models, under multitask and multidimensional settings, with its latent GP inferred with elliptical slice sampling. The paper includes extensive empirical evidence to support FKL's superior performance over common kernels. The paper exposites this method in a clear and understandable manner with a step-by-step process. However, from my perspective it can benefit from some improvements on certain theoretical and empirical aspects. Here are my detailed comments on said aspects. In Section 3.1, the description of the Bochner's theorem is crucial to understanding stationary kernels, but it is slightly inaccurate. Bochner's theorem maps stationary kernels to finite measures, instead of Lebesgue measures (line 78), hence not all spectral measures of stationary kernels have valid spectral densities. For a detailed account of this issue, see Samo and Roberts [1]. The trapezoid rule as a variant of the Darboux sum poses another issue: when the density S(omega) is Lebesgue measurable, the Darboux sum is a good approximator only when S(omega) is continuous almost everywhere (Riemann integrable). However, those issues can be resolved with the fact that a mixture of Gaussian measures is dense in the Banach space of finite measures (see Shen et al.[2]), suggesting Riemann integrable densities are dense for spectral measures. In Section 3.2, the logarithm of the spectral density is modeled by a Gaussian process, but the quadratic mean function of this GP is without proper justification. It is slightly confusing why a parametrized, quadratic mean function is necessary for spectral density estimation on a compact subset of the frequency domain (eq. 3). Section 3.3 explores FKL with multidimensional data, which is modeled with a separable product kernel, rendering it inherently restrictive. Theoretically, separable kernels do not support all multidimensional stationary covariances (one example is the multivariate spectral mixture kernel[3]). For complete support over stationary kernels, it is needed to generalize eq. 3 into a multidimensional setting where omega_i are placed on a Cartesian grid. This generalization represents the entirety of stationary functions, but incurs the curse of dimensionality. While product kernels are commonly used in Gaussian process models, it is worth mentioning the capacity of different model specifications. For inference with FKL, the paper proposes an MCMC based inference scheme using elliptical slice sampling. While the inference is suitable for this task, I think it would benefit from formulating the GP regression for f as Bayesian linear regression instead. The trapzoid rule approximation renders the corresponding Gaussian process degenerate, equivalent to a Bayesian linear regression with trigometric basis expansion. In Section 5.4, comparisons for GP interpolation are made between FKL and standard kernels (RBF, RBF w/ ARD, Matérn). Standard kernels generally do not perform well when the dataset exhibits clear periodicities, for their spectral measures converge around frequency 0. It would be prudent to at least consider spectral mixture kernel (with sparse GP regression) as a candidate for parametric, flexible kernels. On a minor note, the placement of Figure 1 can be improved, and the citation style needs to be more consistent. [1] Samo, Y.-L. K., and Roberts, S.. "Generalized spectral kernels." arXiv preprint arXiv:1506.02236 (2015). [2] Shen, Z., Heinonen, M., and Kaski, S.. (2019). Harmonizable mixture kernels with variational Fourier features. Proceedings of Machine Learning Research, in PMLR 89:3273-3282 [3] Yang, Z., Wilson, A. G., Smola, A., and Song, L.. "A la carte–learning fast kernels." Artificial Intelligence and Statistics. 2015

Reviewer 3



----- Originality ----- The basic idea of this work is not new, but the practical construction is neat and can be of interest to the machine learning community. ----- Quality ----- The technical parts appear correct to me, although I have not been digging into the mathematical details. I enjoy the probabilistic kernel model, since my own impression of highly parametrised approaches for increased expressiveness is that they tend to be very challenging to train. The fact that the proposed approach easily extends to multiple input- and output dimensions is promising, since this is required in many real-world applications. Although the proposed method can describe the entire class of stationary covariance functions (which is a limitation, although of interest for a broad class of problems), it seems to me that the particular interest is functions with periodic behaviour. Therefore, it seems a bit strange to me that no comparison is made to the standard periodic covariance function obtained through the warping u(x)=(cos(x),sin(x)), which have shown promising results in modelling periodic functions elsewhere (see e.g. [1] below). I think that would be the most natural and simplest design choice to capture periodic behaviour. Furthermore, the focus in the one-dimensional comparison with other methods is on extrapolation; apart from the RBF kernel, the interpolation performance seems similar across all methods. In the multiple input case when focus is on interpolation, the performance difference as compared to the Matern kernel decreases - it would be interesting to hear the authors view on whether this improvement is justified by the higher model complexity. Finally, I am wondering about the time scaling properties when solving problems of multiple input dimensions. Intuitively it seems like the numerical integration (Eq. 3) is a bottleneck in these cases, so it would be interesting to know more about it. [1] Ghassemi & Deisenroth, "Analytic long-term forecasting with periodic Gaussian processes", AISTATS 2014. ----- Clarity ----- The paper is very well-written, easy to follow, with clear methodological descriptions. ----- Significance ----- I believe that this paper by itself is significant to the sub-field of GP modelling that focuses on functions with periodic behaviour. Although I am not sure that there are plenty of problems that will benefit from this particular construction, I have a feeling that it has good potential of inspiring more powerful extensions and developments.

[Author Response · NeurIPS 2019]

We thank all the reviewers for their supportive and insightful comments. While kernel learning has now been broadly identified as important for good performance, the vast majority of approaches, while highly useful, focus on parametric methods that do not represent uncertainty over the values of the kernel, can be difficult to train, and difficult to specify inductive biases. Our proposed functional kernel learning (FKL) approach provides a Bayesian nonparametric distribution over kernels, with (a) support for a wide range of kernels; (b) uncertainty representation; (c) easy specification of inductive biases through prior means on the distribution over the spectral density; (d) automatic inference without requiring extensive intervention; (e) natural multidimensional and multi-output generalizations; (f) exhaustive experimental results over a wide range of problems supporting the procedure. Moreover, we want to emphasize that the approach is broadly applicable, and does well on data with and without periodic structure. We would be grateful if reviewers could consider our response in determining their ultimate assessment.

**R2**: *Bayesian Linear Regression:* Thank you for pointing out the theoretically and practically useful connection to Bayesian linear regression with trigonometric basis functions. When we use a transformed Gaussian process prior on the spectral density with a Matern-3/2 kernel, which is mean square continuous, we make the assumption that $S(\omega)$ is a continuous density function – which we will clarify in the camera ready version. However, one can make other choices of covariance function for the GP on the spectrum. To show density amongst spectral measures, we can adapt Theorem 5 of [3] to our setting, noting that the trapezoid rule can be shown to be equivalent to both Riemann and Darboux sums. For discontinuous but finite measures the trapezoid rule will provide an approximator of an underlying stationary kernel on the compact set $[0, \omega_{max}]$, converging as $\omega_{max} \to \infty$ (e.g. as the number of basis functions goes to infinity), where the trapezoids can represent mixtures of Gaussians on the spectral density, and Gaussian mixtures are dense approximations of Riemann integrable densities [3] (by collapsing onto point masses). For continuous spectral densities, we note that in practice the rate of convergence would typically be much faster than a standard Fourier series approximation corresponding to point masses on the spectrum.

*Latent Mean function*: We choose a quadratic mean function for the GP on the spectral density to induce a prior expectation of an RBF kernel (line 115). A major and distinctive advantage of the FKL model is the ability to specify the prior distribution over kernel classes, which can provide a powerful inductive bias. FKL with a quadratic mean will have the inductive biases of an RBF kernel, but has the ability to respond to patterns in the data to learn non-RBF covariance structures. *Higher dimensions*: We will revise the writing of this section to emphasize a) the limitations of product kernels in high dimensions and b) the need to use multivariate FFTs to fully represent multi-dimensional stationary functions. *Comparisons with SM kernels on UCI*: Inspired by your feedback, we ran several experiments with exact product SM kernels, which performed somewhat worse than FKL (and require a lot more manual intervention and careful initialization). The RMSEs for product SM kernels are shown below.

| fertility | concreteslump | servo | machine | yacht | housing | energy |
|---|---|---|---|---|---|---|
| $0.199 \pm 0.038$ | $63.857 \pm 8.111$ | $0.276 \pm 0.117$ | $1.024 \pm 0.181$ | $0.224 \pm 0.083$ | $9.54 \pm 1.668$ | $9.681 \pm 16.62$ |

**R1**: Thanks for your helpful comments. In the camera ready, we will fix the typos and add in-text references to the figures we missed. *1,2*: For non-stationary kernels, we will discuss how we could extend to the multivariate generalized Fourier transform [1,2]. Non-axis aligned methods are also possible with other generalizations of FFT (possibly [3]). *3*: Yes, this means that scale factors are unnecessary.

Spectral Mixture Data

*5*: We will clarify that $\Delta$ is chosen to be no greater than the maximum space between any adjacent points along any dimension of the data. *7*: We standardize and de-mean the number of passengers per month in 1000s. In the camera ready, we will update the figure to be on the count instead. *9*: FKL in its current form is most natural for continuous outputs, but could be adapted (with an appropriate likelihood) for ordinal or categorical data.

**R4**: Thanks for your thoughtful comments. *Time scaling properties in increased dimensions:* We would like to clarify that the product kernel for multiple dimensions separates as a product of one dimensional integration problems allowing the computational complexity to scale linearly with dimension. *Comparisons with periodic warped GPs:* We have tried comparisons to GPs with periodic kernels. Inspired by your feedback, we ran the experiment with associated figure on the left, where the data are drawn from a GP with spectral mixture kernel. We see FKL outperforms a standard GP with a periodic kernel. We also note that many of the UCI datasets do not have quasi-periodic behaviour and we still see FKL outperforming standard kernels; the benefit of FKL is that the kernel is not restricted to a particular functional form and can learn many types of stationary kernels.

**References**:

[1] Samo and Roberts, Generalized Spectral Mixture Kernels, https://arxiv.org/abs/1506.02236
[2] Remes et al, Generalized Spectral Mixture Kernels, NeurIPS, 2017.
[3] Shen, Z., Heinonen, M., and Kaski, S. Harmonizable mixture kernels with variational Fourier features. AISTATS, 2019.


[Meta-Review · NeurIPS 2019]

Reviewers have provided additional comments after the discussion period. I ask the author to include the additional clarifications suggested by the reviewers.